# Extended-spectrum beta-lactamase-producing *Escherichia coli* and *Klebsiella pneumoniae*: insights from a tertiary hospital in Southern Thailand

Chonticha Romyasamit,[1] Phoomjai Sornsenee,[2] Soontara Kawila,[3] Phanvasri Saengsuwan[4]

**ABSTRACT** Broad-spectrum ampicillin-resistant and third-generation cephalosporin-resistant Enterobacteriaceae, particularly *Escherichia coli* and *Klebsiella pneumoniae* that have pathological features in humans, have become a global concern. This study aimed to investigate the prevalence, antimicrobial susceptibility, and molecular genetic features of extended-spectrum beta-lactamase (ESBL)-producing *E. coli* and *K. pneumoniae* isolates in Southern Thailand. Between January and August 2021, samples ($n = 199$) were collected from a tertiary care hospital in Southern Thailand. ESBL and AmpC-lactamase genes were identified using multiplex polymerase chain reaction (PCR). The genetic relationship between ESBL-producing *E. coli* and *K. pneumoniae* was determined using the enterobacterial repetitive intergenic consensus (ERIC) polymerase chain reaction. ESBL-producing *E. coli* and *K. pneumoniae* isolates were mostly collected from catheter urine samples of infected female patients. The ESBL production prevalence was highest in the medical wards ($n = 75$, 37.7%), followed by that in surgical wards ($n = 64$, 32.2%) and operating rooms ($n = 19$, 9.5%). Antimicrobial susceptibility analysis revealed that all isolates were resistant to ampicillin, cefotaxime, ceftazidime, ceftriaxone, and cefuroxime; 79.4% were resistant to ciprofloxacin; and 64.3% were resistant to trimethoprim-sulfamethoxazole. In ESBL-producing *K. pneumoniae* and *E. coli*, $bla_{TEM}$ ($n = 57$, 72.2%) and $bla_{CTX-M}$ ($n = 61$, 50.8%) genes were prominent; however, no $bla_{VEB}$, $bla_{GES}$, or $bla_{PER}$ were found in any of these isolates. Furthermore, only ESBL-producing *K. pneumoniae* had co-harbored $bla_{TEM}$ and $bla_{SHV}$ genes at 11.6%. The ERIC-PCR pattern of multidrug-resistant ESBL-producing strains demonstrated that the isolates were clonally related (95%). Notably, the presence of multidrug-resistant and extremely resistant ESBL producers was 83.4% and 16.6%, respectively. This study highlights the presence of $bla_{TEM}$, $bla_{CTX-M}$, and co-harbored genes in ESBL-producing bacterial isolates from hospitalized patients, which are associated with considerable resistance to beta-lactamase and third-generation cephalosporins.

**IMPORTANCE** We advocate for evidence-based guidelines and antimicrobial stewardship programs to encourage rational and appropriate antibiotic use, ultimately reducing the selection pressure for drug-resistant bacteria and lowering the likelihood of ESBL-producing bacterial infections.

**KEYWORDS** enterobacterial repetitive intergenic consensus polymerase chain reaction, genetic relationship, multidrug-resistant isolates, hospitalized patients, antimicrobial susceptibility analysis

Enterobacteriaceae is a large family of Gram-negative, non-spore-forming, rod-shaped, facultative anaerobes. Several members of this group, including *Escherichia*, *Enterobacter*, *Klebsiella*, *Proteus*, *Citrobacter*, *Serratia*, *Salmonella*, *Shigella*, and *Yersinia*,

Address correspondence to Phanvasri Saengsuwan, sphanvas@medicine.psu.ac.th.

The authors declare no conflict of interest.

See the funding table on p. 11.

cause serious hospital-acquired and community-onset bacterial infections in humans (1). *Escherichia coli* and *Klebsiella* spp. are the most common causative pathogens of infections, especially in countries with poor healthcare systems. *E. coli* is a component of the human and animal gut microbiota but can also be found in water, soil, and vegetation. *Klebsiella* spp. are major opportunistic pathogens that can cause infectious diseases (2, 3).

Extended-spectrum beta-lactamases (ESBLs) that contain *E. coli* and *K. pneumoniae* are major causes of childhood infections and pose significant challenges, such as treatment failure due to multidrug resistance and high morbidity and mortality (4). ESBLs are class A beta-lactamases, a rapidly growing group of hydrolyzable beta-lactamases resistant to oxy-amino cephalosporins [cefotaxime (CTX), ceftazidime, ceftriaxone, cefuroxime, and cefepime] and monobactams (aztreonam) (5). The empirical and symptomatic (without a diagnosis) use of antibiotics in resource-poor settings has increased the incidence of bacterial antibiotic resistance (4, 5). Common infections caused by ESBL-producing bacteria now necessitate multidrug treatments, leading to a greater dependence on carbapenem antibiotics and, consequently, an elevated risk of developing resistance to this antibiotic class (6).

ESBLs are caused by TEM-1, TEM-2, and SHV-1 beta-lactamase mutations. Over 350 natural ESBL variants have been classified into nine distinct structural and evolutionary families based on amino acid sequence comparisons, including Temoniera in Greece beta-lactamases (TEM), sulf-hydryl variable active site (SHV), cefotaximase reference to its preferential hydrolytic activity against CTX, first isolated at Munich (CTX-M), *Pseudomonas* extended resistant (PER), Vietnamese extended-spectrum beta-lactamase (VEB), Guiana extended spectrum (GES), Brazil extended spectrum, named after the Tlahuicas Indians, and oxacillinase (OXA), where the main variants are TEM, SHV, CTX-M, and OXA (7, 8).

In 2017, there were an estimated 197,400 cases of ESBL-producing Enterobacteriaceae among hospitalized patients and 9,100 estimated deaths in the United States (6). Southeast Asia and the Western Pacific Regions have a combined population of 4.3 billion (global population 7.7 billion), which includes two of the most populous countries with heavy antibiotic consumption, namely, China and India. Research suggests that these regions have high rates of antimicrobial resistance due to ESBL-producing bacteria in the pediatric population (9). Forty studies from 11 countries and areas in a meta-analysis identified 2,411 positive and 2,874 negative samples for antibiotic resistance. Furthermore, a high risk of ESBL-producing bacterial infection was associated with previous hospital care, notably intensive care unit stays [pooled odds ratio (OR) 6.5, 95% CI 3.04–13.73] and antibiotic exposure (OR 4.8, 95% CI 2.25–10.27) (9).

ESBL-producing Enterobacteriaceae resistance to antimicrobial agents, especially *E. coli* and *K. pneumoniae*, poses a substantial issue in nosocomial infections and community settings. The widespread use of antibiotics in medical clinics and animal farms has enabled ESBL-producing bacteria to evolve and become increasingly prevalent owing to mutations. The prevalence of ESBL-producing bacteria among inpatients has been investigated; however, these have mostly focused on adult patients (10–13). Consequently, varying prevalence of ESBL-producing bacteria has been established. However, little is known regarding the epidemiology of ESBL-producing variants in Thailand. Moreover, it is critical to provide updated resistance patterns as these might affect the treatment decisions in this region. Thus, this study aims to investigate the prevalence and phenotypic and genotypic characteristics of ESBL-producing *E. coli* and *K. pneumoniae* in Southern Thailand.

## MATERIALS AND METHODS

### Description of participants

#### Bacterial isolation

Bacterial samples ($n$ = 199; *E. coli* = 120, *K. pneumoniae* = 79) and sterile blood were collected as per the hospital records. Catheter urine and tissue samples were collected from individual patients with nosocomial infections at the Songklanagarind Hospital, a 1,000-bed tertiary care hospital in Songkhla province, Southern Thailand. The clinical samples were acquired from patients between January 2021 and August 2021. Each isolate was inoculated onto MacConkey agar supplemented with ceftriaxone (4 mg/L) to screen for ESBL strains. Samples were incubated at 37°C for 24 h. Matrix-assisted laser desorption ionization/time-of-flight mass spectrometry was used to identify and confirm all *E. coli* and *K. pneumoniae* isolates (11). Bacterial characterization and antibiotic susceptibility tests were conducted at the Microbiology Unit, Department of Pathology, Faculty of Medicine, Prince of Songkla University. The isolates were stored at −80°C until further use.

#### Antimicrobial sensitivity testing

Antimicrobial susceptibility tests were performed using the disk diffusion method according to the Clinical and Laboratory Standards Institute guidelines (14). Each disk (Becton Dickinson, Heidelberg, Germany) contained ampicillin (10 µg), colistin (10 µg), ertapenem (ETP, µg), gentamicin (10 µg), imipenem (10 µg), meropenem (MEM, 10 µg), amikacin (AK, 30 µg), CTX (30 µg), ceftazidime (30 µg), ceftriaxone (30 µg), cefuroxime (30 µg), ciprofloxacin (5 µg), trimethoprim-sulfamethoxazole (1.25/23.75 µg), norfloxacin (10 µg), cefoperazone/sulbactam (Sulperazone) (75/30 µg), or piperacillin/tazobactam (Tazocin) (100/10 µg), and fosfomycin (200 µg).

Antimicrobial resistance was then categorized as follows: multidrug-resistant (MDR) strains, isolates resistant to at least one drug in three or more different antimicrobial categories; extensively drug-resistant (XDR) strains, isolates resistant to at least one drug in all but two or fewer antimicrobial categories; and pandrug-resistant (PDR) strains resistant to all classes except colistin (15). Data for this study are available in the National Center for Biotechnology Information (NCBI) BioProjects database (PRJNA984445).

#### Determination of ESBL producers

ESBL production was evaluated using the disk diffusion method according to the Clinical and Laboratory Standards Institute guideline (14), performed with ceftazidime (30 µg), cefotaxime (30 µg), and ceftriaxone (10 µg) (Becton Dickinson). All isolates resistant to at least one of the cephalosporins indicators were interpreted as having a positive ESBL phenotype. *E. coli* American Type Culture Collection 25922 was used as a quality control strain.

#### Bacterial genomic DNA extraction

Bacterial genomic DNA was extracted from all strains according to the manufacturer's instruction (Vivantis Technologies, Sendirian Berhad, Malaysia). A spectrophotometer (Shimadzu UV-1800; Shimadzu, Kyoto, Japan) at an absorbance of 260 nm (A260) was used to measure DNA concentrations. The purity of DNA was determined using the A260:A280 ratio, and DNA quality was assessed using agarose gel electrophoresis.

#### ESBL-producing isolate genotypic characterization

Multiplex polymerase chain reaction (PCR) was performed to screen for the presence of six ESBL-encoding genes: $bla_{TEM}$ (12), $bla_{SHV}$ (12), $bla_{CTX-M}$ (7), $bla_{GES}$ (11), $bla_{VEB}$ (11), and $bla_{PER}$ (11) using specific primers described in Table 1. The PCR reaction mixture contained 1× Taq buffer, 1.5-mM $MgCl_2$, 400-µM deoxynucleotide triphosphates (dNTPs), 0.2-µM forward and reverse primers, 1-U Taq polymerase, and 50-ng/µL DNA template.

**TABLE 1**  List of extended-spectrum beta-lactamase primer sequences used in this study

| Genes | Primer names | Primer sequences (5′–3′) | Product size (bp) | Reference |
|---|---|---|---|---|
| $bla_{TEM}$ | TEM-F | TCGCCGCATACACTATTCTCAGAATGA | 445 | (12) |
| | TEM-R | ACGCTCACCGGCTCCAGATTTAT | | |
| $bla_{SHV}$ | SHV-F | ATGCGTTATATTCGCCTGTG | 747 | |
| | SHV-F | TGCTTTGTTATTCGGGCCAA | | |
| $bla_{CTX-M}$ | CTX-M-F | ACCGCCGATAATTCGCAGAT | 588 | (7) |
| | CTX-M-R | GATATCGTTGGTGGTGCCATAA | | |
| $bla_{GES}$ | GES-F | TAC TGG CAG SGA TCG CTC AC | 838 | (11) |
| | GES-R | TTG TCC GTG CTC AGG ATG AG | | |
| $bla_{VEB}$ | VEB-F | GCC AGA ATA GGA GTA GCA AT | 703 | |
| | VEB-R | TGG ACT CTG CAA CAA ATA CG | | |
| $bla_{PER}$ | PER-F | CTC AGC GCA ATC CCC ACT GT | 851 | |
| | PER-R | TTG GGC TTA GGG CAG AAA GCT | | |

PCR amplification (C1000 Touch; Bio-Rad Laboratories, Hercules, CA, USA) was performed as follows: initial denaturation for 10 min at 95℃; followed by 30 amplification cycles for 30 s at 94℃, 40 s at 60℃, and 90 s at 72℃; and a final extension for 10 min at 72℃. After PCR processing, PCR products were examined using agarose gel electrophoresis. A 100-bp DNA ladder (GeneDireX, München, Germany) was used as a molecular-size marker. The PCR products were then sequenced (1st BASE DNA Sequencing Services, Selangor, Malaysia), and their sequence similarity was determined using the Basic Local Alignment Search Tool on the NCBI database (16). All data sets were submitted to the NCBI Sequence Read Archive (BioProject No. PRJNA984445).

### Enterobacterial repetitive intergenic consensus-PCR genotyping

All isolates were typed by enterobacterial repetitive intergenic consensus (ERIC)-PCR using the protocol described by Versalovic et al. (17) with some modifications. ERIC-1 (5′-ATGTAAGCTCCTGGGGATTCAC-3′) and ERIC-2 (5′-AAGTAAGTGACTGGGGTGAGCG-3′) were used as primers. Each PCR reaction contained 1× Taq buffer, 0.5-µM dNTPs, 1-U/µL Taq DNA polymerase, 3-µM $MgCl_2$, 1-µM forward and reverse primers, and 50-ng/µL template genomic DNA. Amplifications were performed using a C1000 Thermal Cycler (Bio-Rad Laboratories) under the following temperature profile: initial denaturation at 95℃ for 5 min; 35 cycles for 1 min at 95℃, 1 min at 48℃, and 1 min at 72℃; and a final extension at 72℃ for 10 min. The ERIC-PCR products were separated by electrophoresis on a 1% agarose gel using ViSafe green gel stain (0.001%, vol/vol; Vivantis Technologies) and visualized with the Gel Doc XR+ system (Bio-Rad Laboratories).

ERIC-PCR patterns were analyzed using BioNumerics software (version 7.0; Applied Maths, Sint-Martens-Latem, Belgium). A similarity matrix was estimated using Dice's coefficient, and a dendrogram was created based on the unweighted-pair group method with arithmetic averages. ESBL isolates with a similarity coefficient of ≥85% were considered the same genotype (18).

### Statistical analysis

Demographic data are presented as counts with percentage or median values with an interquartile range (IQR). All statistical data were analyzed using SPSS Statistics (version 23; SPSS Inc., Chicago, IL, USA). Independent sample $t$-tests were used to compare continuous variables between groups. Statistical significance was measured using two-tailed tests, and significance was set at a $P$ value of $<0.05$.

## RESULTS

### Study population characteristics

Of the 199 ESBL isolates obtained from patients, 106 (53.3%) and 93 (46.7%) were obtained from women and men, respectively. The age distribution revealed that the

maximum number of ESBL-producing bacteria (*E. coli* and *K. pneumoniae*) was observed in patients aged 65 years and above (50.3%; IQR 59–65 years, range 0–90 years). The highest prevalence of ESBL isolates was found in catheter urine. Of these isolates from catheter urine, 33 (44.8%) were *K. pneumoniae* and 53 (44.2%) were *E. coli*. The medical ward had the highest percentage of ESBL producers ($n = 75$, 37.7%) followed by the surgical ward ($n = 64$, 32.2%), operating room ($n = 19$, 9.5%), and intensive care units ($n = 16$, 8.0%). In addition, the study found ESBL-producing *K. pneumoniae* in the coronavirus disease ward ($n = 1$, 0.5%). There is no significant difference between the sex, age, clinical source, and hospital unit linked with ESBL-producing *E. coli* and *K. pneumoniae* strains (Table 2).

## Antimicrobial susceptibility profiles

The ESBL isolates ($n = 199$) were assessed against different classes of antimicrobial agents using the disc-diffusion method. The antimicrobial susceptibilities of these isolates are listed in Table 3. All ESBL producers were resistant to ampicillin, cefotaxime, ceftazidime, ceftriaxone, and cefuroxime 100%, >40% to aminoglycoside, sulfonamide, and 40.7%, 64.3%, and 79.4% to gentamicin, trimethoprim-sulfamethoxazole, and ciprofloxacin, respectively.

Among ESBL-susceptible isolates, resistance to carbapenem and colistin was high (>85%) compared to sulperazone, piperacillin/tazobactam, and gentamicin. In contrast, the maximum susceptible rate was observed for meropenem and ertapenem (99.5%) followed by amikacin (98.0%), colistin (93.5%), and imipenem (87.4%). In addition, among the total of 199 bacterial isolates, 83.4% (166 of 199) were MDR; the highest

**TABLE 2** Extended-spectrum beta-lactamase-producing *Escherichia coli* and *Klebsiella pneumoniae* distribution from Songklanagarind Hospital clinical isolates between January 2021 and August 2021 ($n = 199$)[a]

| Variables | *E. coli* (N = 120) n (%) | *K. pneumoniae* (N = 79) n (%) | Total number of isolates n (%) | P value |
|---|---|---|---|---|
| Sex | | | | 0.981 |
| Female | 64 (53.3) | 42 (53.2) | 106 (53.3) | |
| Male | 56 (46.7) | 37 (46.8) | 93 (46.7) | |
| Age (years) | | | | 0.470 |
| 0–17 | 4 (3.3) | 6 (7.6) | 10 (5.0) | |
| 18–40 | 10 (8.3) | 9 (11.4) | 19 (9.5) | |
| 41–64 | 43 (35.8) | 27 (34.2) | 70 (35.2) | |
| 65+ | 63 (52.5) | 37 (46.8) | 100 (50.3) | |
| Age (years), median (IQR) | 65 (59–65) | | | |
| Clinical source | | | | 0.467 |
| Blood | 19 (15.6) | 16 (20.3) | 35 (17.6) | |
| Body fluid | 40 (33.3) | 21 (26.6) | 61 (30.7) | |
| Catheter urine | 53 (44.2) | 33 (44.8) | 86 (43.2) | |
| Tissue | 8 (6.7) | 9 (11.4) | 17 (8.5) | |
| Hospital unit | | | | 0.794 |
| Coronavirus disease | 0 | 1 (1.3) | 1 (0.5) | |
| Emergency room | 1 (0.8) | 2 (2.5) | 3 (1.5) | |
| Gynecology ward | 7 (5.8) | 3 (3.8) | 10 (5.0) | |
| Intensive care unit | 10 (8.3) | 6 (7.6) | 16 (8.0) | |
| Medical ward | 45 (37.5) | 30 (38.0) | 75 (37.7) | |
| Operating room | 11 (9.2) | 8 (10.1) | 19 (9.5) | |
| Orthopedic ward | 3 (2.5) | 0 | 3 (1.5) | |
| Patient under investigation | 2 (1.7) | 2 (2.5) | 4 (2.0) | |
| Pediatric ward | 3 (2.5) | 1 (1.3) | 4 (2.0) | |
| Surgical ward | 38 (31.7) | 26 (32.9) | 64 (32.2) | |

[a]$P < 0.05$ considered significant.

**TABLE 3** List of antibiotics used to study ESBL-producing *Escherichia coli* and *Klebsiella pneumoniae* isolates in a tertiary care hospital in southern Thailand (N = 199)[a]

| Antibiotics | | | Number of ESBL-producing isolates (%) | | | | | | | | |
|---|---|---|---|---|---|---|---|---|---|---|---|
| | | | *K. pneumoniae* (n = 79) | | | *E. coli* (n = 120) | | | Total (N = 199) | | |
| Mode of action | Drug classes | Drug names (abbreviation) | R (%) | I (%) | S (%) | R (%) | I (%) | S (%) | R (%) | I (%) | S (%) |
| Targeted β-lactam ring | Aminopenicillin | Ampicillin (AMP) | 79 (100.0) | 4 (5.1) | 0 | 120 (100.0) | 0 | 0 | 199 (100.0) | 0 | 0 |
| | Combinations: piperacillin (β-lactamase) and tazobactam (β-lactamase inhibitors) | Piperacillin/tazobactam (TZP) | 10 (12.7) | 23 (29.1) | 45 (57.0) | 3 (2.5) | 5 (4.2) | 113 (94.2) | 13 (6.5) | 28 (14.1) | 158 (79.4) |
| | Cephalosporin | Cefotaxime (CTX) | 79 (100.0) | 0 | 0 | 120 (100.0) | 0 | 0 | 199 (100.0) | 0 | 0 |
| | | Ceftazidime (CAZ) | 79 (100.0) | 0 | 0 | 120 (100.0) | 0 | 0 | 199 (100.0) | 0 | 0 |
| | | Ceftriaxone (CRO) | 79 (100.0) | 0 | 0 | 120 (100.0) | 0 | 0 | 199 (100.0) | 0 | 0 |
| | | Cefuroxime (CXM) | 79 (100.0) | 0 | 0 | 120 (100.0) | 0 | 0 | 199 (100.0) | 0 | 0 |
| | | Ciprofloxacin (CIP) | 65 (82.3) | 2 (2.5) | 11 (13.9) | 93 (77.5) | 3 (2.5) | 25 (20.8) | 158 (79.4) | 5 (2.5) | 36 (18.1) |
| | | Sulperazone (SCF)[b] | 17 (21.5) | 12 (15.2) | 49 (62.0) | 1 (0.8) | 9 (7.5) | 111 (92.5) | 18 (9.0) | 21 (10.6) | 160 (80.4) |
| | Carbapenems | Imipenem (IMP) | 0 | 0 | 67 (84.8) | 0 | 0 | 107 (89.2) | 0 | 0 | 174 (87.4) |
| | | Meropenem (MEM) | 0 | 1 (1.3) | 78 (98.7) | 0 | 0 | 120 (100.0) | 0 | 1 (0.5) | 198 (99.5) |
| | | Ertapenem (ETP) | 1 (1.3) | 0 | 78 (98.7) | 0 | 0 | 120 (100.0) | 1 (0.5) | 0 | 198 (99.5) |
| | Sulfonamide | Trimethoprim-sulfamethoxazole (SXT) | 63 (79.7) | 0 | 15 (19.0) | 65 (54.2) | 0 | 56 (46.7) | 128 (64.3) | 0 | 71 (35.7) |
| | Phosphonic | Fosfomycin (FOS) | 0 | 0 | 0 | 0 | 0 | 31 (25.8) | 0 | 0 | 31 (16.0) |
| Non-targeted β-lactam ring | Aminoglycosides | Amikacin (AK) | 0 | 1 (1.3) | 78 (98.7) | 3 (2.5) | 0 | 117 (97.5) | 3 (1.5) | 1 (0.5) | 195 (98.0) |
| | | Gentamicin (GM) | 33 (41.8) | 0 | 46 (58.2) | 48 (40.0) | 0 | 72 (60.0) | 81 (40.7) | 0 | 118 (59.3) |
| | Fluoroquinolone | Norfloxacin (NOR) | 21 (26.6) | 5 (6.3) | 6 (7.6) | 40 (33.3) | 0 | 14 (11.7) | 61 (30.7) | 5 (2.5) | 20 (10.1) |
| | Polymyxins | Colistin (CL) | 1 (1.3) | 0 | 73 (92.4) | 0 | 0 | 113 (94.2) | 1 (0.5) | 0 | 186 (93.5) |

[a]CLSI, Clinical and Laboratory Standards Institute; ESBL, extended-spectrum beta-lactamase; I, intermediate; R, resistant; S, susceptible.
[b]No-CLSI interpretative criteria. Interpretative according to cefoperazone/sulbactam in Enterobacteriaceae. See CLSI M100 document guidelines for further detailed recommendations on testing for each agent.

MDR strains were detected in *E. coli* (72.3%, 120 of 199), followed by *K. pneumoniae* isolates (27.7%, 46 of 79). XDR strains were only found in *K. pneumoniae* (100.0%, 33 of 33) with no PDR strains. There was a significant association between the patterns and ESBL producer ($P < 0.05$) (Table 4).

## ESBL genotype detection using PCR

$bla_{CTX-M}$, $bla_{TEM}$, and $bla_{SHV}$ gene presence was determined in all ESBL-producing isolates confirmed by multiplex PCR (Fig. 1). Gel electrophoresis revealed bands positive for at least one ESBL gene in 62 (31.2%) isolates, while 44 (22.1%) were negative for the detected ESBL genes ($bla_{CTX-M}$, $bla_{TEM}$, and $bla_{SHV}$).The most common gene was the $bla_{TEM}$ gene, found in 64 (32.2%) of the isolates, followed by $bla_{CTX-M}$ in 62 (31.2%), and $bla_{SHV}$ in 58 (29.1%), and, 23 (11.6%) had co-harbored the $bla_{TEM}$ and $bla_{SHV}$ genes in only *K. pneumoniae*. Conversely, the $bla_{TEM}$ and $bla_{SHV}$ genes were detected in 51.9%–72.2% of isolates (mainly in *K. pneumoniae*), while $bla_{CTX-M}$ was mostly detected in *E. coli* (50.8%). Furthermore, $bla_{GES}$, $bla_{VEB}$, and $bla_{PER}$ were not identified in any of the ESBL isolates. This causes a significant difference between resistance genes in ESBL-producing isolates ($P < 0.05$) (Table 4). All data sets were submitted to the NCBI. Sequences were as follows: $bla_{TEM}$, MW822683.1; $bla_{CTX-M}$, MW822679; and $bla_{SHV}$ MW822681.

**TABLE 4**  Relationship between the phenotypes and genotypes of antibiotic resistance among extended-spectrum beta-lactamase-producing *Escherichia coli* and *Klebsiella pneumoniae* isolates (N = 199)[a]

| Variables | *K. pneumoniae* (*n* = 79) (%) | *E. coli* (*n* = 120) (%) | Total (%) | *P* value |
|---|---|---|---|---|
| Pattern | | | | 0.000[b] |
| MDR | 46 (27.7) | 120 (72.3) | 166 (83.4) | |
| XDR | 33 (100.0) | 0 | 33 (16.6) | |
| PDR | 0 | 0 | 0 | |
| ERIC genotype | | | | 0.005[b] |
| A | 79 (100.0) | 105 (87.5) | 184 (92.5) | |
| B | 0 | 3 (2.5) | 3 (1.5) | |
| U | 0 | 12 (10.0) | 12 (6.0) | |

[a]ERIC, enterobacterial repetitive intergenic consensus; MDR, multidrug resistant; PDR, pandrug resistant; XDR, extensively drug resistant.
[b]*P* < 0.05 considered significant.

## ERIC-PCR analysis

According to the dendrogram, ERIC-PCR revealed 199 distinct ESBL isolates (Fig. 2). The number of bands varied from one to seven, and ERIC fragment sizes ranged between 100 bp and 1.5 kbp. These were classified into two clusters (A and B) at a similarity level of 95%. The predominant A genotype contained 184 isolates (92.5%), whereas the B genotype contained 3 isolates (1.5%). Twelve isolates (6%) were unambiguously genotyped. MDR and XDR ESBL isolates were distributed among all the scattered patterns. Based on statistical correlation tests, the ERIC pattern among ESBL producers

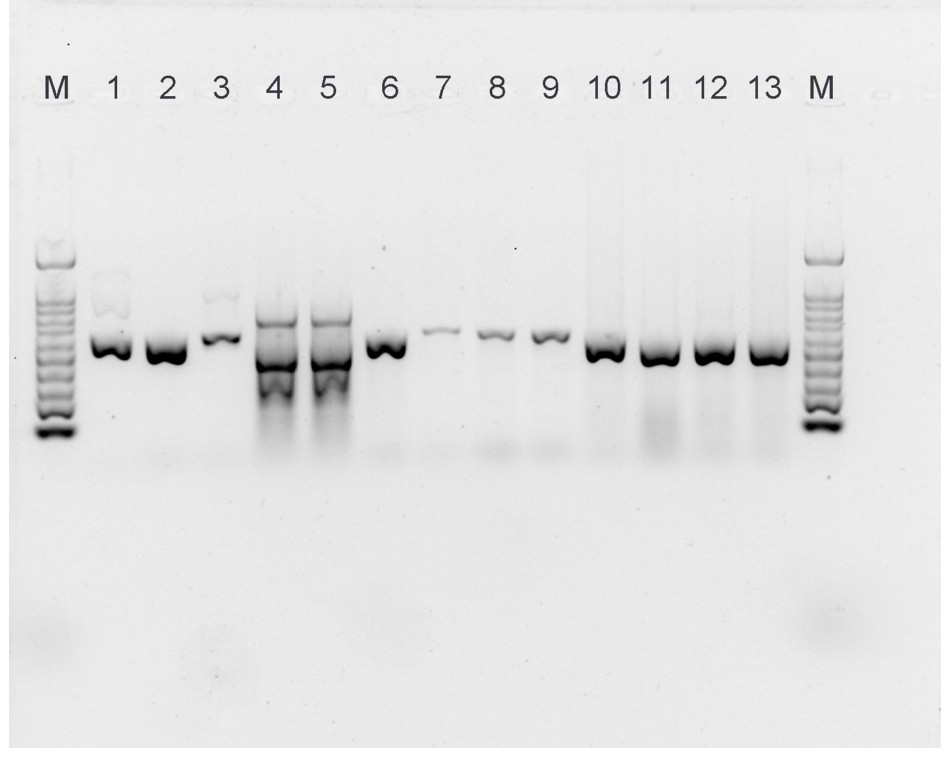

**FIG 1**  Extended-spectrum beta-lactamase (ESBL)-producing *Escherichia coli* and *Klebsiella pneumoniae* isolates in a tertiary care hospital identified using multiplex PCR. Lane M, molecular marker (100–1,000 bp); lanes 1 and 2, ESBL-positive *K. pneumoniae* clinical strains for *bla*CTX-M (588 bp); lane 3, ESBL-positive *K. pneumoniae* clinical strains for *bla*SHV (747 bp); lanes 4 and 5, ESBL-positive *K. pneumoniae* clinical strains for co-harboring *bla*TEM and *bla*SHV (445 and 747 bp); lane 6, positive from ESBL-positive *K. pneumoniae* clinical strains for *bla*TEM (445 bp); lanes 7–9, positive from ESBL-positive *E. coli* clinical strains for *bla*SHV (747 bp); lanes 10–13, ESBL-positive *E. coli* clinical strains for *bla*TEM (445 bp).

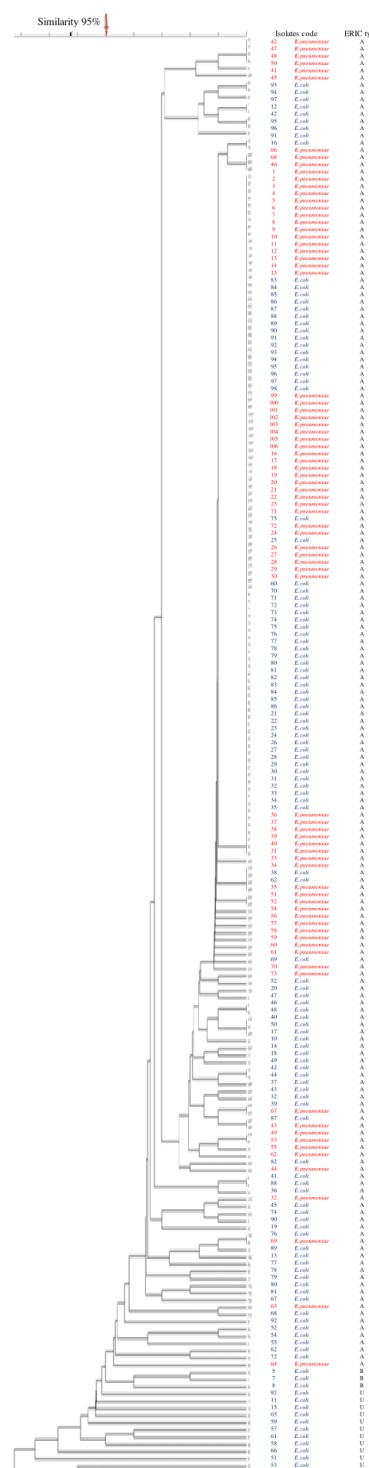

**FIG 2** Dendrogram generated with ERIC-PCR data for ESBL-producing *Escherichia coli* and *Klebsiella pneumoniae* isolates collected from a tertiary care center in Southern Thailand. Similarity of >95% was considered for clustering of isolates. The 199 pulsotypes are designated from A to B and U. ERIC, enterobacterial repetitive intergenic consensus; ESBL, extended-spectrum beta-lactamase; PCR, polymerase chain reaction.

was strongly correlated with the antibiotic resistance patterns (*P* < 0.05). The strains showed high similarity, which may indicate that these isolates are clonal (Table 4).

**TABLE 5** ESBL-encoding gene distribution among *Escherichia coli* and *Klebsiella pneumoniae* isolates ($N = 199$)[b]

| Resistance gene | Number of ESBL-producing isolates (%) | | | P value |
|---|---|---|---|---|
| | *K. pneumoniae* ($N = 79$) | *E. coli* ($N = 120$) | Total ($N = 199$) | |
| | *n* (%) | *n* (%) | *n* (%) | |
| *bla*CTX-M | 1 (1.3) | 61 (50.8) | 62 (31.2) | 0.000[a] |
| *bla*SHV | 41 (51.9) | 17 (14.2) | 58 (29.1) | |
| *bla*TEM | 57 (72.2) | 7 (5.8) | 64 (32.2) | |
| *bla*TEM + *bla*SHV | 23 (29.1) | 0 | 23 (11.6) | |
| *bla*VEB | 0 | 0 | 0 | |
| *bla*GES | 0 | 0 | 0 | |
| *bla*PER | 0 | 0 | 0 | |

[a]$P < 0.05$ considered significant.
[b]ESBL, extended-spectrum beta-lactamase.

## DISCUSSION

To the best of our knowledge, this is the first study to describe ESBL prevalence in clinical specimens from a tertiary care hospital in Southern Thailand. Although catheter urine had the highest incidence of ESBL-producing *E. coli* and *K. pneumoniae*, this could be owing to the greater quantity of urine samples included in this study. Most isolates were acquired from patients in the medical ward (38%). The prevalence of ESBL-producing *E. coli* and *K. pneumoniae* was lower than that reported in other studies with rates of 84.0%, 60.6%, and 55.5% (19–21), which could be because of differences in the frequency of ESBL infection and sample sizes. This suggests that the prevalence of ESBL-producing bacteria varies according to geography and sample size.

Here, ESBL-producing *E. coli* and *K. pneumoniae* isolates showed a high degree of multidrug resistance against some commonly used antimicrobials, namely, ampicillin, cephalosporins, and trimethoprim-sulfamethoxazole. Our findings are consistent with a report from developing countries where ESBL-producing *E. coli* isolated from patients with catheter-associated urinary tract infection showed a higher degree of resistance ranging between 69.6% and 100.0% to most antibiotics, namely, ampicillin, amoxiclav, cephalosporins, and ciprofloxacin (22). Moreover, the overall MDR among all ESBL producers was 83.4%, which is a lower MDR level than studies performed in Ethiopia (93.1%) (23), Nepal (96.8%) (24), and the United Kingdom (97.1%) (25). This may be attributed to the higher number of urine samples collected in this study. The MDR level observed in this study was concerning due to the limited treatment options available for ESBL infections. Consequently, the implementation of robust infection control measures is necessary to mitigate MDR.

Carbapenems are typically the antibiotics of choice for treating severe infections caused by ESBL-producing Enterobacteriaceae (26). In our study, the antibiotic susceptibility of ESBL-producing *E. coli* and *K. pneumoniae* isolates to carbapenems (MEM/ETP) was 99% and 98% to aminoglycosides (AK). Other studies have found similar carbapenem sensitivities (83.0% and 99.3%–100.0%) (27, 28). These results suggest that carbapenems and aminoglycosides should be recommended as first-line drugs for severe infections caused by ESBL-producing Enterobacterales. In addition, colistin remains effective against ESBL infections with a sensitivity of 93%. This result is consistent with previous reports showing that ESBL-producing *E. coli* and *K. pneumoniae* infections are sensitive to ertapenem and colistin (29).

Here, the ESBL-producing *K. pneumoniae* isolates were associated with the presence of *bla*TEM (>70%) (Table 5). This corroborates a previous report that found the *bla*TEM gene among Enterobacteriaceae isolates from patients in Khartoum, Sudan (86%), and Iraq (81%) (30, 31). Furthermore, the *bla*CTX-M-15 gene was the most frequently detected among *E. coli* in a county clinical emergency hospital in Romania (32).

The ERIC-PCR results were organized in two clusters among the 184 ESBL-producing isolates with high similarity (95%). The majority of the 79 ESBL-producing *K. pneumoniae* isolates were clustered in the A genotype, mostly from urine samples, collected from the medical ward. These isolates were resistant to ampicillin, gentamicin, cephalosporin, and co-trimoxazole, consistent with a report from a Ghanaian hospital in South Africa (33). These findings suggest hospital epidemiology stemming from the spread of MDR and XDR among ESBL co-harboring the $bla_{TEM}$ and $bla_{SHV}$ genes. This could be caused by an existing infection with an ESBL-producing bacteria and plasmids conjugated with ESBL genes in this setting, which complicates ESBL infection treatments (34).

Our study presents some limitations. First, the study focused on ESBL-producing isolates from a single center, which may limit the generalizability of the results. It is possible that the prevalence and characteristics of ESBL-producing isolates, as well as the distribution of resistance genes, may differ in other healthcare facilities or regions. Conducting a multicenter study involving multiple hospitals or healthcare facilities would provide a more comprehensive understanding of the situation and increase the external validity of the results. Furthermore, whole-genome sequencing was not performed in this study to determine the CTX variants that are prevalent in our geographic region. In addition, the data were collected retrospectively; thus, patient data such as comorbidities and other clinical information were not considered. In addition, the antimicrobial resistance mechanism and their interactions with virulence factor genes were not investigated. Additional studies will be conducted in the future by performing multilocus sequencing to elucidate the population genetics of ESBL-producing isolates from Thailand, and whole-genome sequences will be used to explore the epidemiological features of ESBL-producing *E. coli* and *K. pneumoniae* isolates in clinical settings. Finally, the utilization of ERIC PCR demonstrates that predominant clones from ESBL-producing *K. pneumoniae* isolates should aid in the control of colonization and transmission in a tertiary care hospital.

Our findings revealed that most of the ESBL-producing *E. coli* and *K. pneumoniae* recovered from patients in Southern Thailand harbored the $bla_{TEM}$, $bla_{CTX-M}$, and $bla_{SHV}$ genes. We clearly demonstrated that ESBL-producing *E. coli* and *K. pneumoniae* epidemics were occurring locally, indicating the need to update the antibiotic therapy strategy to prevent transmission.

## Conclusions

We report the high prevalence of MDR/XDR ESBL-producing *E. coli* and *K. pneumoniae* clinical isolates in Southern Thailand. The presence of resistance genes, such as CTX, SHV, TEM, and co-harbored genes indicates the genetic diversity and potential for the dissemination of drug resistance among these isolates. The identification of genetically closely related strains suggests the endemic spread of specific ESBL-producing isolates in the region. This information is crucial for genotyping and understanding the transmission patterns of drug-resistant pathogens. The application of molecular tools, combined with improved antibiotic stewardship and a "one health" approach, can help combat antimicrobial resistance, reduce the burden of these infections on hospitalized patients, and refine clinical treatment regimens in Southern Thailand. This encourages the logical and appropriate use of antibiotics via antimicrobial stewardship programs. By implementing evidence-based guidelines, healthcare practitioners should reduce the selection pressure for drug-resistant bacteria and decrease the likelihood of ESBL-producing bacterial infections.

## ACKNOWLEDGMENTS

The authors gratefully acknowledge Editage for English language editing.

This study was supported by a grant from the Faculty of Medicine, Prince of Songkla University, with grant number REC63-557-4-7.

C.R. and P.S.: Conceptualization and Data curation; S.K.: Investigation and Data curation; P.S.: Funding acquisition, Data curation, and Writing–original draft preparation, review, and editing. The manuscript, as submitted, has been reviewed and approved by all authors.

## AUTHOR AFFILIATIONS

¹School of Allied Health Sciences, Walailak University, Nakhon Si Thammarat, Thailand

²Department of Family Medicine and Preventive Medicine, Faculty of Medicine, Prince of Songkla University, Hat Yai, Songkhla, Thailand

³Microbiology Unit, Department of Pathology, Faculty of Medicine, Prince of Songkla University, Hat Yai, Songkhla, Thailand

⁴Department of Biomedical Sciences and Biomedical Engineering, Faculty of Medicine, Prince of Songkla University, Hat Yai, Songkhla, Thailand

## AUTHOR ORCIDs

Chonticha Romyasamit http://orcid.org/0000-0003-1403-1628
Phoomjai Sornsenee http://orcid.org/0000-0001-6992-033X
Soontara Kawila http://orcid.org/0009-0007-0122-796X
Phanvasri Saengsuwan http://orcid.org/0000-0002-9982-1524

## FUNDING

| Funder | Grant(s) | Author(s) |
| --- | --- | --- |
| Faculty of Medicine, Prince of Songkla University (Faculty of Medicine) | 63-557-4-7 | Phanvasri Saengsuwan |

## ETHICS APPROVAL

This study was approved by the ethics committees of the Faculty of Medicine, Prince of Songkla University (REC63-557-4-7) and was performed according to the Declaration of Helsinki. Because the samples were collected as part of standard diagnostic care, informed consent was not required.

## ADDITIONAL FILES

The following material is available online.

Open Peer Review

**PEER REVIEW HISTORY (review-history.pdf).** An accounting of the reviewer comments and feedback.

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
