## [Reviewer comments · Microbiology Spectrum]

Microbiology Spectrum

Extended-spectrum beta-lactamase-producing *Escherichia coli* and *Klebsiella pneumoniae*: Insights from a tertiary hospital in Southern Thailand

Chonticha Chonticha Romyasamit, Phoomjai Sornseneee, Soontara Kawila, and Phanvasri Saengsuwan

Corresponding Author(s): Phanvasri Saengsuwan, Prince of Songkla University Faculty of Medicine

Review Timeline:

Submission Date:	January 23, 2024
Editorial Decision:	February 29, 2024
Revision Received:	April 1, 2024
Accepted:	April 12, 2024

Editor: John Atack

Reviewer(s): Disclosure of reviewer identity is with reference to reviewer comments included in decision letter(s). The following individuals involved in review of your submission have agreed to reveal their identity: Joseph Atia Ayariga (Reviewer #3)

Transaction Report:

DOI: <https://doi.org/10.1128/spectrum.00213-24>

Re: Spectrum00213-24 (Extended-spectrum beta-lactamase-producing *Escherichia coli* and *Klebsiella pneumoniae*: Insights from a tertiary hospital in Southern Thailand)

Dear Miss Phanvasri -- Saengsuwan:

Thank you for the privilege of reviewing your work. Below you will find my comments, instructions from the Spectrum editorial office, and the reviewer comments.

Please do not feel it necessary to respond to Reviewer #2 comments - I have included the comments here as their general comments relating to language and presentation will need to be addressed, but this reviewer was unprofessional in their comments and attitude, and you do not need to respond to specific comments.

Revision Guidelines

Sincerely,
John Attack
Editor
Microbiology Spectrum

Reviewer #1 (Comments for the Author):

Review of Romyasamit et al

The authors in this paper surveyed 199 *E. coli* and *K. pneumoniae* 199 isolates from patients for their resistance to various antibiotics and the presence of antibiotic resistance genes.

Notes:

Introduction lines 82-86. All the different classes of beta-lactamases are listed by their acronyms, eg. OXA, SHV etc. But what is not listed is what they stand for, ie. Oxacillinase (OXA). What is not listed is the class B type, metallo beta-lactamases in the introduction. The authors should note that beta-lactamases have four classes, and OXA is class D. This needs to be made clearer.

Results line 236-7. The sentence reads as though all isolates contained all three bands, but I'm sure the authors mean that all 199 isolates were positive for at least one band, not all three. Otherwise, the rest of the analysis does not make sense.

In the materials and methods section, it is mentioned that "ESBL isolates with a similarity coefficient {greater than or equal to} 85% were considered the same genotype (20)." How many of the 199 isolates were considered the same genotype?

Discussion line 263-265: Place the reference next to the reported percentage, rather than at the end of the sentence.

It's good that the authors acknowledge the limitations of the study. They should also note that there's a strong chance that many of the isolates are clonal, hence the fact that the ERIC-PCR result shows that nearly all the isolates are from group A (and 100% of the *K. pneumoniae* ones were). The strong possibility that many of the isolates are clonal needs to be mentioned here (lines 302-316).

Figures:

The labelling for figure one needs to be improved. State in the legend which lanes have which samples and remove the written text from the gel as it is very confusing. Also state what size each band should be in the figure legend.

Figure 2: It is very hard to read figure 2. It looks like there is text at the end of each line on the dendrogram, but it is impossible to read it. Either remove the text or find a way to identify the isolates in an easier way (ie. Different colours for *E. coli* or *K. pneumoniae*). Also, could a gel of the ERIC-PCR be included in the figure?

For tables 4 and 5, state which test you are using to get a p-value (student t-test, odds ratio?)

Overall, the paper reads well and the authors do analyse the data they have presented in a clear and concise manner. The figures need to be improved for publication quality (see above). It is good to know that most of the isolates are carbapenem sensitive, probably owing to the lack of blaPER alleles identified.

One other point, the authors state that they sequenced the alleles. But I did not read any analysis on whether all isolates had the same type of CTX-M allele or TEM type. Were they identical or were different allele types of the resistance genes found? A more detailed discussion on that point should be included.

Reviewer #2 (Comments for the Author):

Abstract: Line 30-32: The description provided is unclear; kindly stick to ESBL-EC and KP or AMPC-EC and KP. Also, the statement seems ambiguous; could you please clarify or rewrite it for better clarity.

Furthermore, I noticed that you later introduced AMPC genes or AMPC beta-lactamase genes alongside ESBL genes. However, I am unsure why you would describe them as AmpC-lactamase genes, as I do not believe this is a description that exists.

A similar issue arises in line 56: "which are associated with considerable resistance to beta-lactamase and third generation cephalosporins". Is this an oversight in writing, or do the authors need to review the facts before writing them up? Resistance to beta-lactamase, really?

Keywords: There are too many keywords and many of them are unrequired. I question whether "ERIC PCR" should be included as a keyword for this paper, for example?

Introduction: Could the authors ensure that the introduction section includes sufficient citations for the statements where they are currently missing?

Line 72-74: Once more, referencing my earlier statement, I would like to address a few concerns: based on my statement earlier. Is this: "Extended-spectrum β -lactamases (ESBLs) that produce *E. coli* and *K. pneumoniae* are major causes of childhood infections and pose significant challenges, such as....." an oversight in writing, or do the authors need to review the facts before writing them up?

Additionally, why start sentences with abbreviations (Check lines 72 and 82).

Also; Lines 72-81 are so uncoordinated. The paragraph is so painful to read.

Line 82: Really now: "ESBLs are caused by TEM-1, TEM-2, and SHV-1 β -lactamase mutations". How is this possible?

Again: ESBL-producing Enterobacteriaceae resistance to antimicrobial agents, especially *E. coli* and *K. pneumoniae*, poses a substantial issue in nosocomial infections and community settings. What is this supposed to mean?

Reviewer #4 (Comments for the Author):

Overview of manuscript: In this paper, the authors examined bacteria previously isolated from hospital patient samples (from a hospital in Southern Thailand) to determine the prevalence of ESBL strains in this locality. They also analyzed their resistance to a few non-B-lactam drugs and used PCR to determine fingerprints for the strains as well as which resistance genes were present. One limitation of the study was that all samples were from the same hospital and may not be representative of the region. However, this study does provide some information as to the prevalence ESBL strains in this area.

Specific comments:

Line 72: rephrase - ESBLs do not produce *E. coli* and *K. pneumoniae*

Line 82: provide a brief description/explanation of these 3 mutations

Lines 161 and 178: the "buffer" needs to be defined, meaning its composition needs to be reported

Lines 178-179: provide the final concentrations of the components (like was done in lines 161-163) rather than volumes and stock concentrations; in addition, it is implied that "empty space" in the reaction is filled with DI water, so this does not need to be stated

Lines 207-232, 286, and potentially elsewhere: Many of the % reported in the manuscript do not match the % reported in the accompanying table; some appear to be variations in rounding and/or rounding errors. However, all of these discrepancies should be corrected, and if any rounding is done, it should be consistent (i.e., to the same decimal point) throughout the manuscript. In addition, in Line 207, these two % add to greater than 100%, so a math error has been made and should be corrected.

Lines 208-209: "maximum number of ESBL-producing bacteria" - it is unclear what exactly is meant by this. Does this mean that all strains identified could be accounted for in the 60+ population and repeats of them were found in younger populations? Does it mean that the greatest variety of strains from a single patient was found in someone over 60? In addition, it is unclear why 60 was chosen as the cut-off when this is not an age cut-off in the table of data provided.

Line 211: "Of these isolates" is vague. Does this refer to isolates from catheter urine?

Lines 211-212: 3 figures for % are provided for only 2 things, so there appears to be an extra number inserted

Lines 216-217 seem to be a repeat of what was stated a few sentences earlier

Lines 223-225: it is not clear why parentheses are used here

Line 224: ciprofloxacin data is not reported in the table. This should be corrected.

Lines 226-227: I don't follow how the prevalence of ESBL-SUSCEPTIBLE isolates was high. In addition, the range of rates for the "other antimicrobials" should be included so readers can judge the comparison for themselves.

Line 232: PDR-PA strains are not mentioned anywhere in Table 5, so there is no reason to reference this table here.

Line 235 and elsewhere: blaCTX-M seems to have about 3 different names throughout the manuscript, tables, and figures - be consistent throughout. If these are referring to different variations, then that needs to be specified and the differences (and their significance) explained.

Line 247: briefly explain ERIC PCR either in the Introduction or here as all readers who are interested in resistance may not be versed in this technique

Line 248: "therefore" should be replaced with "and"

Line 251: explain what is meant by "unambiguously genotyped"

Lines 253-254: explain how this conclusion was reached

Figure 1: whatever is at the ends of the dendrogram branches is not legible; in addition, the figure caption needs to explain what A, B, and U are. It would also be informative for many readers to include an ERIC gel from some small portion of these isolates.

Figure 2: label ladder bands with bp sizes; remove line around the text boxes on the gel; it is not clear what the labels on the gel correspond to, especially because there seems to be many more band sizes than labels; in the caption, for Lanes 1-6, there are 5 things listed corresponding to the 6 lanes, so it is not clear what corresponds to which lanes; presumably lanes 4 and 5 are the lanes showing "co-harboring" yet the bands in those lanes are not the same size as bands in any other lanes despite these supposedly being present singly in other isolates; similarly, there are 3 things listed for lanes 7-13, so it is again not clear what corresponds to each lane.

Table 2: it is not clear what the p-values correspond to/measure

Table 2: in light of lines 259-260, the number of specimens/samples in each category should be stated in the table. This will also

help readers see how often multiple isolates were found in a single sample.
Tables 4 and 5: it is not clear what the p-values correspond to/measure

Review of Romyasamit *et al*

The authors in this paper surveyed 199 *E. coli* and *K. pneumoniae* 199 isolates from patients for their resistance to various antibiotics and the presence of antibiotic resistance genes.

Notes:

Introduction lines 82-86. All the different classes of beta-lactamases are listed by their acronyms, eg. OXA, SHV etc. But what is not listed is what they stand for, ie. Oxacillinase (OXA). What is not listed is the class B type, metallo beta-lactamases in the introduction. The authors should note that beta-lactamases have four classes, and OXA is class D. This needs to be made clearer.

Results line 236-7. The sentence reads as though all isolates contained all three bands, but I'm sure the authors mean that all 199 isolates were positive for *at least one* band, not all three. Otherwise, the rest of the analysis does not make sense.

In the materials and methods section, it is mentioned that "*ESBL isolates with a similarity coefficient $\geq 85\%$ were considered the same genotype (20).*" How many of the 199 isolates were considered the same genotype?

Discussion line 263-265: Place the reference next to the reported percentage, rather than at the end of the sentence.

It's good that the authors acknowledge the limitations of the study. They should also note that there's a strong chance that many of the isolates are clonal, hence the fact that the ERIC-PCR result shows that nearly all the isolates are from group A (and 100% of the *K. pneumoniae* ones were). The strong possibility that many of the isolates are clonal needs to be mentioned here (lines 302-316).

Figures:

The labelling for figure one needs to be improved. State in the legend which lanes have which samples and remove the written text from the gel as it is very confusing. Also state what size each band should be in the figure legend.

Figure 2: It is very hard to read figure 2. It looks like there is text at the end of each line on the dendrogram, but it is impossible to read it. Either remove the text or find a way to identify the isolates in an easier way (ie. Different colours for *E. coli* or *K. pneumoniae*). Also, could a gel of the ERIC-PCR be included in the figure?

For tables 4 and 5, state which test you are using to get a p-value (student t-test, odds ratio?)

Overall, the paper reads well and the authors do analyse the data they have presented in a clear and concise manner. The figures need to be improved for publication quality (see above). It is good to know that most of the isolates are carbapenem sensitive, probably owing to the lack of *bla_{PER}* alleles identified.

One other point, the authors state that they sequenced the alleles. But I did not read any analysis on whether all isolates had the same type of CTX-M allele or TEM type. Were they identical or were different allele types of the resistance genes found? A more detailed discussion on that point should be included.

March 25, 2024

Dr. Christina Cuomo
Editor in Chief
Microbiology Spectrum

Dear Editor:

We wish to re-submit the manuscript titled “Extended-spectrum beta-lactamase-producing *Escherichia coli* and *Klebsiella pneumoniae*: Insights from a tertiary hospital in Southern Thailand.” The manuscript ID is 00213-24.

We thank you and the reviewers for your thoughtful suggestions and insights. The manuscript has benefited from these insightful suggestions. I look forward to working with you and the reviewers to move this manuscript closer to publication in the *Microbiology Spectrum*.

The manuscript has been rechecked and the necessary changes have been made in accordance with the reviewers’ suggestions. The responses to all comments have been prepared and attached herewith/given below.

Thank you for your consideration. I look forward to hearing from you.

Sincerely,
Miss Phanvasri Saengsuwan
Scientist- Profession level, Department of Biomedical Sciences and Biomedical Engineering,
Faculty of Medicine, Prince of Songkla University 90110, Thailand
Phone number: 66 74 45 1181
Email Address: sphanvas@medicine.psu.ac.th

March 29th, 2024

Department of Biomedical Sciences and Biomedical Engineering,
Faculty of Medicine, Prince of Songkla University 90110, Thailand
Phone No: 66 74 45 1181
Email Address: sphanvas@medicine.psu.ac.th

Dear Editor:

We thank the reviewers for their generous comments on the manuscript. We have edited the manuscript in line with these comments.

All of the codes we wrote are available, and we have incorporated multiple links to the appropriate code repositories throughout the paper.

We believe that the manuscript is now suitable for publication in *Microbiology Spectrum*.

Miss Phanvasri Saengsuwan
Scientist- Profession level, Department of Biomedical Sciences and Biomedical Engineering,
Faculty of Medicine, Prince of Songkla University 90110, Thailand

On behalf of all authors.

Review 1:

Notes:

Introduction lines 82-86. All the different classes of beta-lactamases are listed by their acronyms, eg. OXA, SHV etc. But what is not listed is what they stand for, ie. Oxacillinase (OXA). What is not listed is the class B type, metallo beta-lactamases in the introduction. The authors should note that beta-lactamases have four classes, and OXA is class D. This needs to be made clearer.

Response: Thank you for pointing this out. We have thoroughly reviewed the addressing of beta-lactamases in the Introduction section, referencing Bajpai T, Pandey M, Varma M, Bhatambare GS. "Prevalence of TEM, SHV, and CTX-M Beta-Lactamase genes in the urinary isolates of a tertiary care hospital," published in Avicenna J Med in 2017; 7:12–16, and Sharma J, Sharma M, Ray P. "Detection of TEM & SHV genes in Escherichia coli & Klebsiella pneumoniae isolates in a tertiary care hospital from India," found in the Indian J Med Res in 2010; 132:332–36. We have defined each abbreviation of beta-lactamase in line 84-89 of the revised manuscript.

Results line 236-7. The sentence reads as though all isolates contained all three bands, but I'm sure the authors mean that all 199 isolates were positive for at least one band, not all three. Otherwise, the rest of the analysis does not make sense.

Response: We have edited and highlighted the text in lines 242-244 of the revised manuscript.

In the materials and methods section, it is mentioned that "ESBL isolates with a similarity coefficient {greater than or equal to} 85% were considered the same genotype (20)." How many of the 199 isolates were considered the same genotype?

Response: A dendrogram was constructed using the unweighted pair group method with arithmetic mean. This grouped the isolates into clusters with greater than 85% similarity, with the scale indicating the percentage of similarity. We have incorporated these critical details into the manuscript. 184 (92.5%) were considered the same genotype A, 3 (1.5%) were considered B type, and unique type were 12 (6.0%). We have edited Table 2 and other relevant details.

Discussion line 263-265: Place the reference next to the reported percentage, rather than at the end of the sentence.

Response: Thank you. As suggested, we have placed the reference next to the reported percentage in line 272 and highlighted the sentence in the Discussion section.

It's good that the authors acknowledge the limitations of the study. They should also note that there's a strong chance that many of the isolates are clonal, hence the fact that the ERIC-PCR result shows that nearly all the isolates are from group A (and 100% of the *K. pneumoniae* ones were). The strong possibility that many of the isolates are clonal needs to be mentioned here (lines 302-316).

Response: Thank you for your insight and comment. Accordingly, we have expanded the necessary details in the revised manuscript in lines 326–328.

Figures:

The labelling for figure one needs to be improved. State in the legend which lanes have which samples and remove the written text from the gel as it is very confusing. Also state what size each band should be in the figure legend.

Response: Thank you for pointing this out. We have edited Figure 2 and included the necessary details.

Figure 2: It is very hard to read figure 2. It looks like there is text at the end of each line on the dendrogram, but it is impossible to read it. Either remove the text or find a way to identify the isolates

in an easier way (ie. Different colours for *E. coli* or *K. pneumoniae*). Also, could a gel of the ERIC-PCR be included in the figure?

Response: We have removed the blurred text, and regarding the concerns about the 300 dpi resolution of the figures, our images were generated using the BioNumerics 7.0 software at the maximum resolution available in dpi. Additionally, we have included a code of ESBL isolates (Figure 2) and all data were submitted to the NCBI (BioProject No. PRJNA984445).

We appreciate your attention to detail. The sample of ERIC-PCR profiles presented in below were analyzed with BioNumerics software, employing Pearson's correlation coefficient to evaluate the similarity among band patterns. Subsequently, a dendrogram was constructed using the unweighted pair group method with arithmetic mean. This grouped the isolates into clusters with greater than 85% similarity, with the scale indicating the percentage of similarity. We have incorporated these critical details into the manuscript.

We have a representative gel image for ERIC-PCR from the Bio-Rad Gel Doc XR Imaging System for capturing agarose gels. Lane M represents molecular marker (100bp), lane 1–16 represents samples showing characteristic ERIC banding pattern.

For tables 4 and 5, state which test you are using to get a p-value (student t-test, odds ratio?)

Response: We have provided independent sample t-tests to compare the values of continuous variables between groups. Statistical significance was measured using two-tailed tests, and significance was set at $P < 0.05$ in line no. 249-250 on page no. 6.

Overall, the paper reads well and the authors do analyse the data they have presented in a clear and concise manner. The figures need to be improved for publication quality (see above). It is good to know that most of the isolates are carbapenem sensitive, probably owing to the lack of *bla*_{PER} alleles identified.

Response: We appreciate your insightful point. We have edited the details, as suggested.

One other point, the authors state that they sequenced the alleles. But I did not read any analysis on whether all isolates had the same type of CTX-M allele or TEM type. Were they identical or were different allele types of resistance genes found? A more detailed discussion on that point should be included.

*Response: CTX-M and TEM alleles have identical allele types. *bla*_{CTX-M} was detected in 1 (1.3%) *K. pneumoniae* and 61 (50.8%) *E. coli* isolates. Table 4 shows that *bla*_{TEM} was detected in 57*

(72.2%) K. pneumoniae isolates and 7 (5.8) E. coli isolates. We have added the necessary details to the Discussion section.

Reviewer #2 (Comments for the Author):

Abstract: Line 30-32: The description provided is unclear; kindly stick to ESBL-EC and KP or AMPC-EC and KP. Also, the statement seems ambiguous; could you please clarify or rewrite it for better clarity.

Response: We have revised the abstract to meet the word count limit.

Furthermore, I noticed that you later introduced AMPC genes or AMPC beta-lactamase genes alongside ESBL genes. However, I am unsure why you would describe them as AmpC-lactamase genes, as I do not believe this is a description that exists.

Response: Thank you for bringing this to our attention. We acknowledge the observation.

A similar issue arises in line 56: "which are associated with considerable resistance to beta-lactamase and third generation cephalosporins". Is this an oversight in writing, or do the authors need to review the facts before writing them up? Resistance to beta-lactamase, really?

Response: We have read numerous publications that are not overly optimistic before writing.

Keywords: There are too many keywords and many of them are unrequired. I question whether "ERIC PCR" should be included as a keyword for this paper, for example?

Response: We have included a keyword for the genotypic features of ESBL-producing K. pneumoniae and E. coli in Southern Thailand to an ERIC PCR.

Introduction: Could the authors ensure that the introduction section includes sufficient citations for the statements where they are currently missing?

Response: We have included missing citations in the introduction section. This includes citations was in revised the manuscript accordingly.

Line 72-74: Once more, referencing my earlier statement, I would like to address a few concerns: based on my statement earlier. Is this: "Extended-spectrum β -lactamases (ESBLs) that produce E. coli and K. pneumoniae are major causes of childhood infections and pose significant challenges, such as....." an oversight in writing, or do the authors need to review the facts before writing them up?

Response: We have revised it as follows: ESBLs are encoded by genes found on large plasmids that share genes for antimicrobial resistance with other pathogens. ESBLs are frequently transmitted by plasmids and can thereby be distributed among hospitalized patients, driving their spread across regions (Paterson and Bonomo, 2005; Paterson, 2006). The increasing emergence of ESBL-producing pathogens has been documented worldwide and varies among countries (Quan et al., 2017; Abayneh et al., 2018; Mineau et al., 2018; Kettani Halabi et al., 2021).

Additionally, why start sentences with abbreviations (Check lines 72 and 82).

Response: We apologize for this error. We have made the required corrections and tried our best to avoid the use of abbreviations at the start of sentences. However, in some instances (For eg. ESBL) they were unavoidable, as using the full term would just increase the word count.

Also; Lines 72-81 are so uncoordinated. The paragraph is so painful to read.

Response: We have adjusted the paragraph to improve readability.

Line 82: Really now: "ESBLs are caused by TEM-1, TEM-2, and SHV-1 β -lactamase mutations". How is this possible?

Response: We found evidence supporting the statement that ESBLs can be caused by TEM-1, TEM-2, and SHV-1 β -lactamase mutations upon cross-referencing this information with reference no. 7, Bajpai T, Pandey M, Varma M, Bhatambare GS. "Prevalence of TEM, SHV, and CTX-M Beta-Lactamase genes in the urinary isolates of a tertiary care hospital," Avicenna J Med. 2017; 7:12–16, and reference no. 8, Sharma J, Sharma M, Ray P. "Detection of TEM & SHV genes in Escherichia coli & Klebsiella pneumoniae isolates in a tertiary care hospital from India," Indian J Med Res. 2010; 132:332–36

Again: ESBL-producing Enterobacteriaceae resistance to antimicrobial agents, especially *E. coli* and *K. pneumoniae*, poses a substantial issue in nosocomial infections and community settings. What is this supposed to mean?

Response: Thank you for your comments. We would like to highlight that numerous research papers and studies support the idea that ESBL-producing Enterobacteriaceae resistance poses a significant issue in both nosocomial infections and infections acquired in community settings. Below are a few examples of relevant papers:

1. "Extended-spectrum β -lactamases: a clinical update" by Paterson DL and Bonomo RA, published in Clinical Microbiology Reviews in 2005, discusses the emergence and impact of ESBL-producing bacteria in healthcare settings.

2. "Epidemiology and outcome of nosocomial and community-onset bloodstream infection caused by Enterobacteriaceae producing extended-spectrum beta-lactamase" by Kang CI et al., published in the Journal of Antimicrobial Chemotherapy in 2005, highlights the clinical significance of ESBL-producing Enterobacteriaceae infections.

3. "A review of the epidemiology and treatment of infections caused by extended-spectrum beta-lactamase-producing Enterobacteriaceae" by Pitout JDD and Laupland KB, published in Clinical Microbiology and Infection in 2008, provides a comprehensive overview of the challenges posed by ESBL-producing Enterobacteriaceae.

These examples are just a few among many studies that discuss the issue of ESBL-producing Enterobacteriaceae resistance in both nosocomial and community settings.

Reviewer #4 (Comments for the Author):

Overview of manuscript: In this paper, the authors examined bacteria previously isolated from hospital patient samples (from a hospital in Southern Thailand) to determine the prevalence of ESBL strains in this locality. They also analyzed their resistance to a few non-B-lactam drugs and used PCR to determine fingerprints for the strains as well as which resistance genes were present. One limitation of the study was that all samples were from the same hospital and may not be representative of the region. However, this study does provide some information as to the prevalence of ESBL strains in this area.

Specific comments:

Line 72: rephrase - ESBLs do not produce *E. coli* and *K. pneumoniae*

Response: We have edited and highlighted the text in line 75.

Line 82: provide a brief description/explanation of these 3 mutations.

Response: We would like to emphasize that CTX-M-producing E. coli and K. pneumoniae are increasingly implicated in urinary tract infections, as highlighted by Arpin et al. (2009). ESBL production by pathogenic E. coli and K. pneumoniae isolates has been documented by numerous investigators worldwide, with prevalence rates steadily rising in various healthcare settings, ranging from 6% to 88% (Saedii et al., 2017; Zongo et al., 2015; Iroha et al., 2010). Additionally, it is noteworthy that the blaTEM and blaCTX-M genes are among the most commonly encountered ESBL-producing genes, as reported by Ahmad and Khalil (2019).

Lines 161 and 178: the "buffer" needs to be defined, meaning its composition needs to be reported.

Response: The buffer used is Taq buffer. We have defined and highlighted the "Taq" in the Methods section.

Lines 178-179: provide the final concentrations of the components (like was done in lines 161-163) rather than volumes and stock concentrations; in addition, it is implied that "empty space" in the reaction is filled with DI water, so this does not need to be stated.

Response: Thank you for your valuable comments and suggestions. We have incorporated the necessary details, as suggested.

Lines 207-232, 286, and potentially elsewhere: Many of the % reported in the manuscript do not match the % reported in the accompanying table; some appear to be variations in rounding and/or rounding errors. However, all of these discrepancies should be corrected, and if any rounding is done, it should be consistent (i.e., to the same decimal point) throughout the manuscript. In addition, in Line 207, these two % add to greater than 100%, so a math error has been made and should be corrected.

Response: We apologize for the error and thank you for your valuable comments and suggestions. In the revised file, we have rechecked these results and revised the manuscript accordingly.

Lines 208-209: "maximum number of ESBL-producing bacteria" - it is unclear what exactly is meant by this. Does this mean that all strains identified could be accounted for in the 60+ population and repeats of them were found in younger populations? Does it mean that the greatest variety of strains from a single patient was found in someone over 60? In addition, it is unclear why 60 was chosen as the cut-off when this is not an age cut-off in the table of data provided.

Response: Thank you for pointing this out. We have edited the necessary details. Our study primarily focuses on ESBL infections. We did not employ specific sampling methods but instead collected all samples from clinical specimens of patients admitted to the hospital for at least 48 hours. Our findings align with previous research indicating that individuals over the age of 60 exhibit the highest prevalence of ESBL-producing isolates, specifically E. coli and K. pneumoniae. Numerous studies have corroborated that advanced age, typically over 60 years, is a significant risk factor for severe infection with ESBL producers (Hongsuwan et al., 2014; Heytens et al., 2017; Kettani Halabi et al., 2021). We appreciate your feedback, and we will consider these comments for future studies.

Line 211: "Of these isolates" is vague. Does this refer to isolates from catheter urine?

Response: Yes, it does. We have added the highlighted text in line 216-217.

Lines 211-212: 3 figures for % are provided for only 2 things, so there appears to be an extra number inserted.

Response: We have made the necessary corrections.

Lines 216-217 seem to be a repeat of what was stated a few sentences earlier.

Response: We have deleted the repetitive sentences in the result section.

Lines 223-225: it is not clear why parentheses are used here.

Response: Thank you for pointing this out. We have deleted parentheses.

Line 224: ciprofloxacin data is not reported in the table. This should be corrected.

Response: We have defined the resistance rate in Table 3.

Lines 226-227: I don't follow how the prevalence of ESBL-SUSCEPTIBLE isolates was high. In addition, the range of rates for the "other antimicrobials" should be included so readers can judge the comparison for themselves.

Response: We have edited lines 231-232.

Line 232: PDR-PA strains are not mentioned anywhere in Table 5, so there is no reason to reference this table here.

Response: We agree with your insightful observation. We deleted the words " PDR-PA " and added the necessary details, as suggested.

Line 235 and elsewhere: *bla*CTX-M seems to have about 3 different names throughout the manuscript, tables, and figures - be consistent throughout. If these are referring to different variations, then that needs to be specified and the differences (and their significance) explained.

Response: We have rechecked and revised the manuscript, tables, and Figure 1. The correct format is blaCTX-M.

Line 247: briefly explain ERIC PCR either in the Introduction or here as all readers who are interested in resistance may not be versed in this technique.

Response: We have elaborated here on the enterobacterial repetitive intergenic consensus-polymerase chain reaction (ERIC-PCR) as a simplified typing method for hospital-based epidemiology, as discussed by Shahi et al. (2020). These strategies aimed at identifying dominant clones are crucial for controlling colonization and transmission within a tertiary care setting.

Line 248: "therefore" should be replaced with "and"

Response: Thank you for your suggestion. We have replaced it with "and" in the highlighted text in line no. 255.

Line 251: explain what is meant by "unambiguously genotyped"

Response: In our study, we have identified 12 unique traits of ESBL isolates, indicating the circulation of these strains within Songklanagarind Hospital prior to the current study. These findings imply a hospital epidemiology characterized by the dissemination of multidrug-resistant (MDR) strains and endemic ESBL genes. It is plausible that these bacteria play a role in the propagation of antibiotic resistance and resistance genes through horizontal gene transfer.

Lines 253-254: explain how this conclusion was reached.

Response: Based on our study findings, the emergence of multidrug-resistant (MDR) and ESBL-producing E. coli and K. pneumoniae isolates with elevated rates of antibiotic resistance to commonly used antibiotics, coupled with the heightened prevalence of major gene types such as blaTEM, as well as other genes including blaCTX and blaSHV, presents significant concerns for physicians and microbiologists alike. Regular monitoring of antibiotic susceptibility and associated genes is essential to guide appropriate antibiotic use for treating major pathogens like E. coli and K. pneumoniae in community hospitals and healthcare facilities.

Figure 1: whatever is at the ends of the dendrogram branches is not legible; in addition, the figure caption needs to explain what A, B, and U are. It would also be informative for many readers to include an ERIC gel from some small portion of these isolates.

Response: We agree with your insightful observation. As a result, we have added the necessary details in Figure 2.

Figure 2: label ladder bands with bp sizes; remove line around the text boxes on the gel; it is not clear what the labels on the gel correspond to, especially because there seems to be many more band sizes than labels; in the caption, for Lanes 1-6, there are 5 things listed corresponding to the 6 lanes, so it is not clear what corresponds to which lanes; presumably lanes 4 and 5 are the lanes showing "co-harboring" yet the bands in those lanes are not the same size as bands in any other lanes despite these supposedly being present singly in other isolates; similarly, there are 3 things listed for lanes 7-13, so it is again not clear what corresponds to each lane.

Response: We agree with your perceptive comment. We have edited the figure details, as suggested.

Table 2: it is not clear what the p-values correspond to/measure.

Response: The sex representation value of 0.981 is not significant for E. coli and K. pneumoniae isolates. For the age group, clinical source, and infectious unit, the P values of 0.470, 0.467, and 0.794 were likewise not significant. We have added the necessary details in line 219-220.

Table 2: in light of lines 259-260, the number of specimens/samples in each category should be stated in the table. This will also help readers see how often multiple isolates were found in a single sample.

Response: We have defined individual isolates that were collected from individual patients with nosocomial infections in the Materials & Methods section on page 4.

Tables 4 and 5: it is not clear what the p-values correspond to/measure

Response: We have revised the P values in Table 4 and P values from Table 5 represent the correlation between resistance genes and ESBL producing isolates.

Re: Spectrum00213-24R1 (Extended-spectrum beta-lactamase-producing *Escherichia coli* and *Klebsiella pneumoniae*: Insights from a tertiary hospital in Southern Thailand)

Dear Miss Phanvasri -- Saengsuwan:

Your manuscript has been accepted, and I am forwarding it to the ASM production staff for publication. Your paper will first be checked to make sure all elements meet the technical requirements. ASM staff will contact you if anything needs to be revised before copyediting and production can begin. Otherwise, you will be notified when your proofs are ready to be viewed.

Sincerely,
John Attack
Editor
Microbiology Spectrum